ⓐ | **Open Peer Review** | Environmental Microbiology | Research Article

# Tracking *Vibrio*: population dynamics and ecology of *Vibrio parahaemolyticus* and *V. vulnificus* in an Alabama estuary

Blair H. Morrison,[1,2,3] Jessica L. Jones,[3] Brian Dzwonkowski,[1,2] Jeffrey W. Krause[1,2]

**ABSTRACT**  *Vibrio* is a genus of halophilic, gram-negative bacteria found in estuaries around the globe. Integral parts of coastal cultures often involve contact with vectors of pathogenic *Vibrio* spp. (e.g., consuming raw shellfish). High rates of mortality from certain *Vibrio* spp. infections demonstrate the need for an improved understanding of *Vibrio* spp. dynamics in estuarine regions. Our study assessed meteorological, hydrographic, and biological correlates of *Vibrio parahaemolyticus* and *V. vulnificus* at 10 sites in the Eastern Mississippi Sound System (EMSS) from April to October 2019. During the sampling period, median abundances of *V. parahaemolyticus* and *V. vulnificus* were 2.31 log MPN/L and 2.90 log MPN/L, respectively. *Vibrio* spp. dynamics were largely driven by site-based variation, with sites closest to freshwater inputs having the highest abundances. The E-W wind scalar, which affects Ekman transport, was a novel *Vibrio* spp. correlate observed. A potential salinity effect on bacterial-particle associations was identified, where *V. vulnificus* was associated with larger particles in conditions outside of their optimal salinity. Additionally, *V. vulnificus* abundances were correlated to those of harmful algal species that did not dominate community chlorophyll. Correlates from this study may be used to inform the next iteration of regionally predictive *Vibrio* models and may lend additional insight to *Vibrio* spp. ecology in similar systems.

**IMPORTANCE**  *Vibrio* spp. are bacteria found in estuaries worldwide; some species can cause illness and infections in humans. Relationships between *Vibrio* spp. abundance, salinity, and temperature are well documented, but correlations to other environmental parameters are less understood. This study identifies unique correlates (e.g., E-W wind scalar and harmful algal species) that could potentially inform the next iteration of predictive *Vibrio* models for the EMSS region. Additionally, these correlates may allow existing environmental monitoring efforts to be leveraged in providing data inputs for future Vibrio risk models. An observed correlation between salinity and *V. vulnificus*/particle-size associations suggests that predicted environmental changes may affect the abundance of *Vibrio* spp. in certain reservoirs, which may alter which vectors present the greatest vibrio risk.

**KEYWORDS**  *Vibrio*, *Vibrio parahaemolyticus*, *Vibrio vulnificus*, environmental microbiology, estuarine, bacteria-phytoplankton interactions, marine microbial ecology, phytoplankton

Address correspondence to Blair H. Morrison, bmorrison@mobilebaynep.com.

The authors declare no conflict of interest.

See the funding table on p. 19.

Coastal communities, like those along the northern Gulf of Mexico (GOM), are intrinsically linked to aquatic ecosystem services. Integral aspects of coastal culture—eating raw shellfish (i.e., oysters), fishing, and recreation—can involve contact with *Vibrio* spp. vectors. *Vibrio* is a cosmopolitan genus of halophilic, gram-negative bacteria found in marine and estuarine waters (1–4) and contains about a dozen human pathogenic species (1, 4–6). In the northern GOM, *Vibrio parahaemolyticus* and *V. vulnificus* are the pathogenic species of greatest concern (3, 7). The United States

Center for Disease Control and Prevention estimates that over 80,000 cases of vibrio-related illness occur nationally each year (8), with raw oyster consumption being largely responsible for enteric cases of infection (vibriosis). *V. parahaemolyticus* causes the most enteric vibriosis cases, whereas *V. vulnificus* has the greatest mortality rate (9). Most individuals recover fully from vibriosis, but 25% of those infected by *V. vulnificus* die (8), and mortality rates are higher in those with certain pre-existing medical conditions (7, 9). These outcomes demonstrate the need for an improved understanding of potential ecological drivers of *V. parahaemolyticus* and *V. vulnificus* populations.

Coastal *Vibrio* spp. abundances correlate to hydrographic factors (10), such as temperature, e.g., ≥20 ° C (11), and salinity, e.g., 5–20 ppt and 15–36 ppt for *V. vulnificus* (12–15) and *V. parahaemolyticus* (12, 16, 17), respectively. Estuarine conditions also affect vibrios. Fluvial input decreases salinity and pH and increases turbidity (reducing water-column irradiance) (18). Freshwater discharge also alters water-column stratification and concentrations of dissolved nutrients (nitrogen, phosphorus, and silicic acid), and can diminish the euphotic zone depth (18–20). Within salinity-favorable areas for vibrios, higher turbidity from fluvial input positively affects *Vibrio* spp. abundances (21, 22). However, the relative importance of temperature, salinity, turbidity, nutrient concentration, and euphotic zone depth in shaping *Vibrio* spp. abundance is variable (23), and site specificity complicates applying generalized trends. This variation highlights the need for regional studies to investigate complex interactions among physical, hydrographic, and ecological drivers of multiple *Vibrio* species (24).

Beyond abiotic factors, many bacteria (including *Vibrio* spp.) associate with organic detritus and planktonic organisms (9, 25–29). *Vibrio* spp. are known to associate with chitinous organisms and phytoplankton aggregates (26, 29–31); such associations may allow *Vibrio* spp. to use "phycospheres" - small regions of concentrated organics created by phytoplankton exudates (32, 33). Similarly, associations may be strong with harmful bloom-forming dinoflagellate species (23, 34), due to their high production of organic exudates. Ties among phytoplankton and vibrios can be so tightly coupled that Constantin de Magny et al. (35) used lagged chlorophyll-a anomalies to predict abundances of *V. cholerae*; however, this trend may not be universal in estuaries (36).

The Eastern Mississippi Sound System (EMSS) is an estuary in the northern GOM, which includes western Mobile Bay, coastal embayments (Grand Bay, Portersville Bay, and Fowl River Bay), eastern Mississippi Sound, and barrier islands (Petit Bois and Dauphin Island) along its southern boundary (Fig. 1). Under its micro-tidal regime (37), most freshwater draining from the ~1.1×10$^5$ km$^2$ Mobile Bay watershed exits through its southern boundary (i.e., Main Pass, 38, 39) while ~25% to 33% discharges through the Pass-aux-Herons into the EMSS (40). During northeast wind conditions, disproportionately higher discharge can flow through Pass-aux-Herons (31). Varying freshwater input and meteorological forcings create complex biophysical gradients that may affect *Vibrio* spp. and planktonic assemblages in the EMSS (40–44). Thus, the EMSS may serve as a model system for understanding the physical forcing mechanisms of *Vibrio* spp. dynamics in shallow, stratified, and microtidal subtropical estuarine systems.

Although vibrio-plankton associations have been investigated in other estuarine systems, these relationships have not been reported in the EMSS. Several dinoflagellate species with documented *Vibrio* spp. correlations elsewhere are known constituents of the regional phytoplankton regime. Holiday (46) determined that salinity, dissolved organic phosphorus, and dissolved organic nitrogen are the most important structuring factors to the phytoplankton community in Mobile Bay and the EMSS; accordingly, we hypothesize that the *Vibrio* spp. of interest will co-occur with phytoplankton communities that share overlapping hydrographic requirements.

*Vibrio* spp. dynamics in the EMSS are an important issue for public health. Physical and biotic correlates of *Vibrio* spp. have been largely understudied in the GOM; therefore, even short-term and regional studies can uncover valuable correlates that may be useful for risk prediction. Previous work has examined temporal dynamics of *V. cholerae* abundances in the water column, sediments, and oysters (47), but there is still much

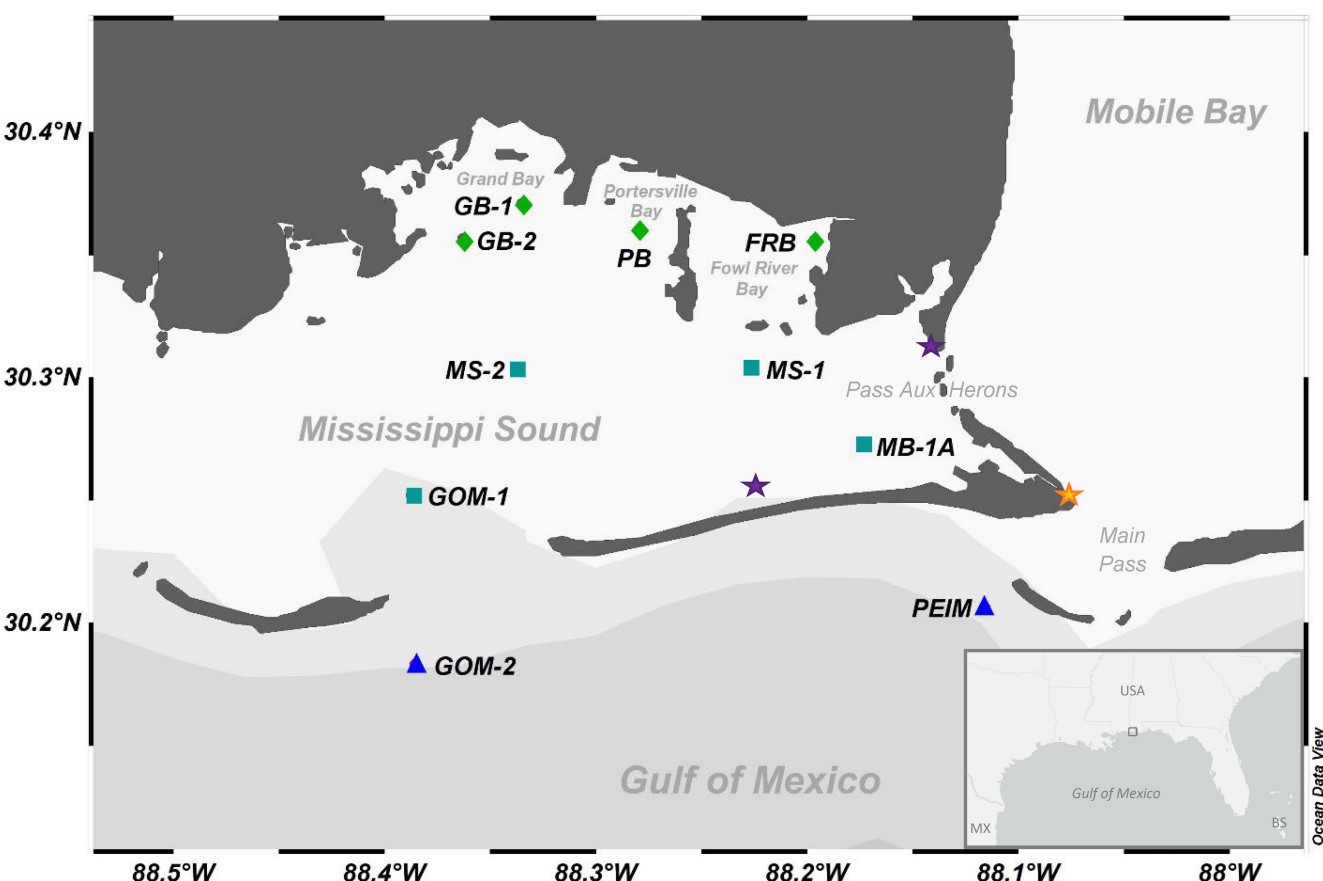

**FIG 1** Sampling sites in the EMSS. Symbols represent ADEM water quality monitoring sites, sampled monthly from April to October 2019. These sites are roughly grouped into three sections: coastal bays (green diamonds), sound (teal squares), and barrier islands (dark blue triangles). The orange star represents the Dauphin Island (DI) ARCOS station used for reference meteorological and hydrographic data. Purple stars represent the locations of two other ARCOS stations that were considered for reference points. Map was created using Ocean Data View software (45). The study area (small gray box) is shown in the context of the larger northern Gulf of Mexico region on the map in the lower right corner.

unknown about the regional population dynamics of other *Vibrio* spp. Our study aims to better assess meteorological, hydrographic, and biological correlates of *V. parahaemolyticus* and *V. vulnificus* in the EMSS. Due to the integrative scope of this study, our data may offer new pathways to examine when modeling *Vibrio* spp. abundances.

## RESULTS

### Meteorology, hydrography, and nutrients

The 2019 mean wind direction was south-southeast (Fig. 2a), with lower average wind speeds occurring during the summer months (May–September) (Fig. 2b). Between April and October 2019, windspeeds ranged from 1.22 m/s to 15.41 m/s, and in contrast to the dominant annual wind direction, northeast winds were the most frequent (Fig. S1).

The Dauphin Island (DI) Alabama Real-time Coastal Observing System (ARCOS) meteorological station recorded 137.7 cm of rain in 2019 (historical average ~150–170 cm). The most intense rainfall events occurred in the months of April–July during our study period (Fig. S2a). The pulses of rainfall align with increases in the freshwater discharge from the Mobile and Tensaw rivers (Fig. S2b)—major sources of freshwater to northern Mobile Bay. Freshwater discharge remained above 900 m$^3$/s for each river from January to May, which were the highest rates of the year.

The near-continuous DI ARCOS station recorded water temperature, salinity, and turbidity (24 hour average) ranges between 15.8 and 33.3°C; 1.9 and 32.9 ppt; and

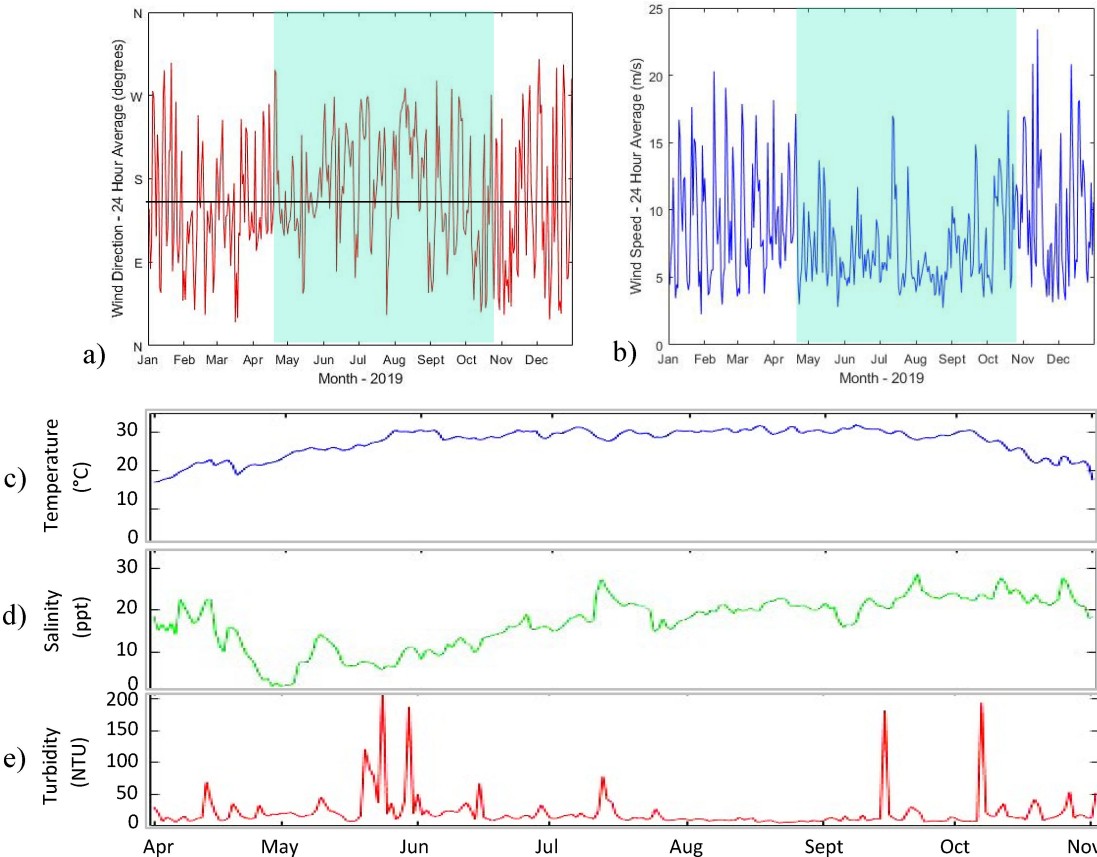

**FIG 2** DI ARCOS station meteorological and hydrographic data during 2019 (48). (a) Daily average wind direction for 2019; the black line is the mean wind direction (SSE). (b) Daily average wind speed (m/s). Shaded blue box in a, b shows the study sampling period. (c) Water temperature (°C), (d) salinity (ppt), and (e) turbidity (NTU) were collected continuously during the study period.

5.1 and 121 NTU, respectively, during the sampling period (Fig. 2c through e). Among the discrete *in situ* samples, the ranges were smaller (temperature 20.0–31.0°C, salinity 4.8–32.7 ppt, and turbidity 0.4–39.4 NTU; Fig. 3) and, unsurprisingly, failed to capture some of the more extreme events where conditions were unsafe to sample. Temperature fluctuated seasonally, with the lowest values at the beginning and end of the sampling window (April–May and October). Salinity displayed a seasonality with lowest values in the early sampling months, consistent with freshwater inputs earlier in the hydrographic year (Fig. 2d). Turbidity did not display clear seasonal trends in the ARCOS or *in situ* data sets, although salinity and turbidity measurements taken at sampling stations were negatively correlated with each other ($\rho_s = -0.595$, $P < 0.001$). This negative correlation was mirrored in the continuous water quality DI ARCOS station; high turbidity events were typically preceded (within a 3-day period) by notable drops in salinity.

Many samples (72% of $NO_x$ samples and 70% of DRP samples) yielded nutrient concentrations at or below the limit of detection (LOD) for the methods used in this study. Sites with greater marine influence (GOM-2, GOM-1, and PEIM) tended to have higher $NO_x$ concentrations (>0.04 mg/L) than near-shore sites throughout the sampling period; this counterintuitive result is consistent with the high local submarine discharge of $NO_x$-rich groundwater (49). Dissolved reactive phosphorus (DRP) did not yield any clear trends across sites (Fig. S3). From May to October, ammonia levels consistently exceeded the maximum concentration of quantitation (0.09 mg/L) across all sites. The methodology used is the regulatory standard across all state waters; therefore, many assays are optimized for analysis in fresh waters. Standard analysis for ammonia was too sensitive (upper LOD too low) to detect variations in the estuarine environment,

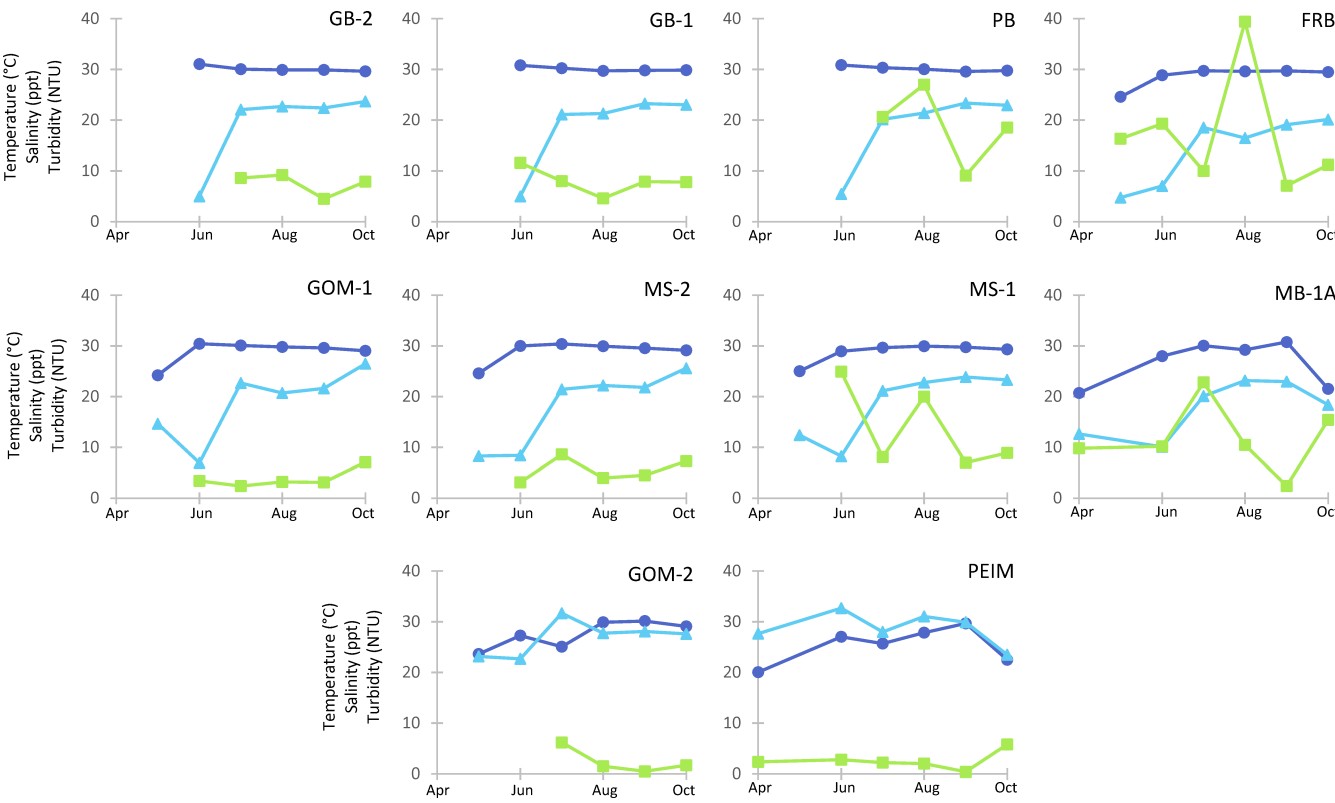

**FIG 3** Temperature (°C, dark blue circles), salinity (ppt, light blue triangles), and turbidity (NTU, green squares) trends among *in situ* sampling sites (plot name) each month. Plots are arranged by site region (top row = coastal bays, middle row = central sound, and bottom row = barrier island) and from west to east (i.e., left to right). Values reported reflect the average parameter value recorded within the euphotic zone at each site.

whereas analyses for $NO_x$ and DRP were not sensitive enough (lower LOD too high). Future studies may wish to conduct additional nutrient analyses using methodologies specifically designed for estuarine and marine samples.

## Biological data sets

### Chlorophyll-a and harmful algal abundance and diversity

The range in observed chlorophyll-a concentrations was <1–7.4 µg/L during the sampling period. There were no significant correlations between chlorophyll-a concentrations and temperature or salinity among sites (Fig. S4a through c). Harmful algal abundances ranged from 100 (LOD) to 48,000 cells/L.

Thirty-three species (among 18 genera) of potentially harmful algae were identified and enumerated during our study period, with dinoflagellates having the greatest diversity. Some species were only recorded once (e.g., *Diplosalis lenticula* and *Katodinium glaucum*), whereas others were recorded throughout the sampling period (*Protoperidineum* spp., *Ceratium hircus*, *Prorocentrum scutellum*, etc.); many species were only found within specific salinity conditions (Fig. S5). Densities exceeding 15,000 cells/L for *Akashiwo sanguinea*, *Prorocentrum micans*, and *Pseudo-nitzschia* spp. were reported in June at site GB-2, August at site MS-2, and September at site PEIM, respectively.

### Vibrio abundances and distribution among particle sizes

*Vibrio* spp. abundances fluctuated throughout the sampling period and among sites. The highest mean summed abundances were observed at FRB, MB-1A, PB, GB-1, and MS-1 (Fig. 4). *V. parahaemolyticus* abundances (averaged from replicate MPNs and summed

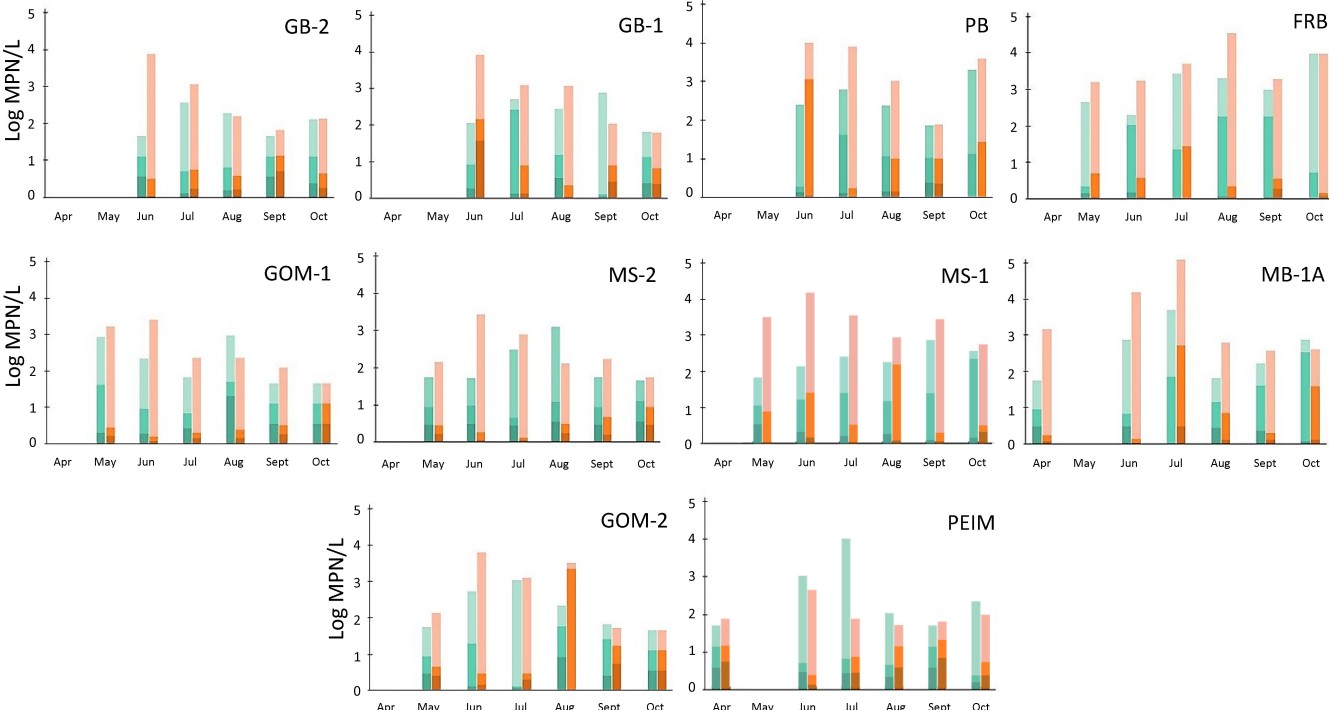

**FIG 4** *Vibrio* spp. abundances [log (base 10) MPN/L] at each site throughout the sampling period. Plots are arranged by site region (top row = coastal bays, middle row = central sound, and bottom row = barrier island) and from west to east (i.e., left to right). *Vibrio parahaemolyticus* abundances are plotted in teal, and *Vibrio vulnificus* abundances are plotted in orange. Shading denotes vibrios associated with particle size fractions: >35 µm (darkest shade), 35–5 µm (medium shade), and <5 µm (lightest shade). Missing columns (April and May) indicate months where vibrios were not able to be sampled at respective sites.

across size fractions) ranged from 1.65 to 3.98 log MPN/L (median 2.38). *V. vulnificus* abundances varied between 1.65 and 5.09 log MPN/L (median 2.85).

On average, 30%–50% of total *Vibrio* spp. in any sample were associated with ≥5 µm particles (Table 1). *V. vulnificus* was primarily associated with smaller particles in lower salinities (median, 11 ppt) and larger particles in higher salinities (median, 22 ppt; Fig. 5a); this was a significant difference in relation to hydrographic parameters (Kruskal-Wallis test: H = 4.13, P = 0.04). A similar relationship was not observed for *V. parahaemolyticus*. Turbidity was not significantly correlated with particle-size association for either species (Fig. 5b).

## Statistical evaluation

### Correlation analyses

Summed *Vibrio* spp. abundances did not significantly correlate with raw wind data, except for summed *V. parahaemolyticus* and raw wind direction ($\rho_s$ = 0.28, *P* = 0.03). However, when compared with the wind components (N-S and E-W scalars), both *V. vulnificus* and *V. parahaemolyticus* abundances displayed a significant negative correlation with the E-W wind scalar (Table 2). The correlation between the E-W wind scalar and *V. vulnificus* abundance was stronger than that of the scalar and *V. parahaemolyticus* abundance. Winds from the west (east) were correlated to an increase (decrease) in *Vibrio* spp. abundance. When integrating over longer time scales, these patterns became weaker over an 8 hour period but were still significant over a 24 hour timeframe.

*Vibrio* spp. abundances correlated with hydrography. There was no significant correlation to temperature for either species in our sampling period, as (expectedly) these months had optimal temperature conditions for *Vibrio* spp. There was a significant negative correlation between *V. vulnificus* and salinity across all size fractions and summed abundances (Table 3); however, this was not observed for *V. parahaemolyticus*

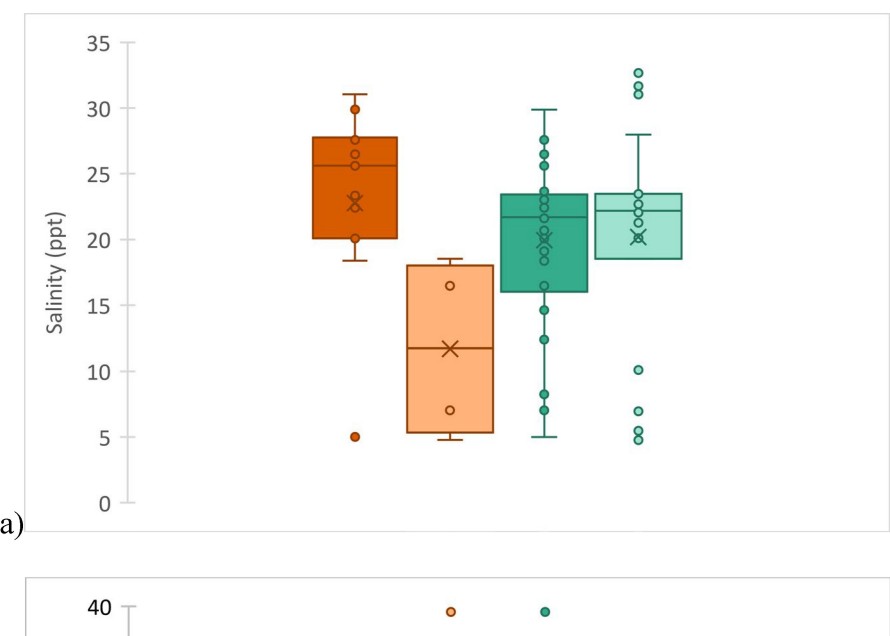

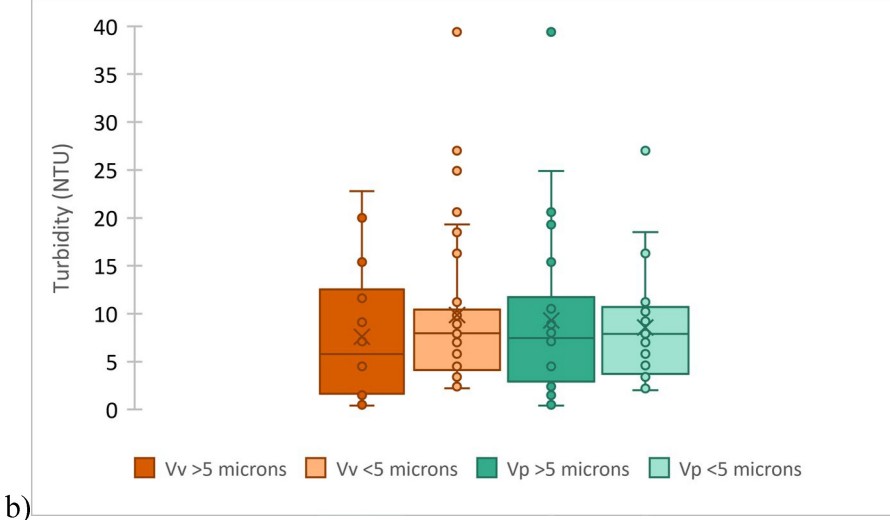

**FIG 5** Box plot of *Vibrio vulnificus* (Vv) and *Vibrio parahaemolyticus* (Vp) associations with size fractions of particles with respect to salinity (a) and turbidity (b).

(see also Fig. 4). Both species' abundances were positively correlated with turbidity. Combined *V. vulnificus* abundances were negatively correlated with $NO_x$. Summed *V. parahaemolyticus* and *V. vulnificus* abundances in the <5 µm size fraction were positively correlated to alkalinity. All vibrio abundances, except *V. parahaemolyticus* in the >35 µm size fraction were negatively correlated to euphotic zone depth.

Vibrio spp. abundances varied with harmful algal genera (Table 4). *V. vulnificus* abundances had significant positive correlations to the abundances of dinoflagellates *A. sanguinea* and *Heterocapsa* spp. but significant negative correlations with the abundances of *Prorocentrum* spp. (dinoflagellate) and *Pseudo-nitzschia* spp. (diatom). *V. parahaemolyticus* abundances only had a significant negative correlation with

**TABLE 1** Average proportion and abundance (±standard error) of *Vibrio* spp. associated with particles at each size fraction (summation of data in Fig. 4)

| Size | *V. parahaemolyticus* | | *V. vulnificus* | |
|---|---|---|---|---|
| Fraction | Average proportion | Average abundance (log MPN/L) | Average proportion | Average abundance (log MPN/L) |
| >35 µm | 0.16 ± 0.02 | 1.38 ± 0.10 | 0.12 ± 0.02 | 1.57 ± 0.15 |
| 35–5 µm | 0.35 ± 0.03 | 1.81 ± 0.17 | 0.21 ± 0.02 | 2.03 ± 0.22 |
| <5 µm | 0.49 ± 0.03 | 2.02 ± 0.19 | 0.67 ± 0.03 | 2.63 ± 0.24 |

**TABLE 2** Spearman's ρ (P values[a]) between summed *V. vulnificus* and *V. parahaemolyticus* abundances and wind scalars[b]

| | *In situ* | *In situ* | 8 hour average | 8 hour average | 24 hour average | 24 hour average |
|---|---|---|---|---|---|---|
| Species | N-S scalar | E-W scalar | N-S scalar | E-W scalar | N-S scalar | E-W scalar |
| *V. vulnificus* | −0.09 (0.50) | −0.30 (**0.03**) | 0.02 (0.86) | −0.26 (0.05)[c] | −0.09 (0.48) | −0.38 (**<0.01**) |
| *V. parahaemolyticus* | 0.06 (0.64) | −0.26 (**0.05**) | 0.20 (0.14) | −0.22 (0.10) | 0.05 (0.73) | −0.33 (**0.01**) |

[a]Significant P values (P < 0.05) are bolded.
[b]In situ refers to the wind scalars during the time of sampling. The 8 hour average refers to winds occurring during an 8 hour period prior to sampling. The 24 hour average refers to winds occurring during a 24 hour period prior to sampling.
[c]8 hour E-W correlation with V. vulnificus is not significant (i.e., rounded down to 0.05).

*Pseudo-nitzschia* spp. These relationships mirror correlations with salinity and turbidity (Tables 3 and 5), potentially indicating community associations with freshwater input (Fig. S6).

Phytoplankton species that correlated with vibrios generally did not dominate community abundances. The genera that correlated with *Vibrio* spp. (*Akashiwo, Heterocapsa, Prorocentrum*, and *Pseudo-nitzschia*), on average, comprised 28% (*A. sanguinea*) to 45% (*Prorocentrum* spp.) of the total phytoplankton cell counts in their respective samples. These genera were the dominant members of the quantified phytoplankton community in fewer than 40% of all samples (*Heterocapsa* spp.–13% of samples; *A. sanguinea*–22% of samples; *Pseudo-nitzschia* spp.–33% of samples; and *Prorocentrum* spp.–38% of samples).

### Linear mixed effects (LME) models

Linear mixed-effects models were created for summed *Vibrio* spp. abundances. Almost all variation in *V. vulnificus* abundances was attributed to site, whereas about 41% of variation in *V. parahaemolyticus* was attributed to site (Table 6). Accounting for the variance attributed to each of the environmental factors retained in the *V. parahaemolyticus* best-fit model (Tables 6 and 7), salinity, N-S wind component, DRP, and ammonia were significant predictors of abundance. Doing the same for the *V. vulnificus* best-fit

**TABLE 3** Spearman's ρ (P values[a]) for hydrographic variables, *V. parahaemolyticus* abundances, and *V. vulnificus* abundances.

| | *V. parahaemolyticus* | | | | *V. vulnificus* | | | |
|---|---|---|---|---|---|---|---|---|
| | >35 µm | 35–5 µm | <5 µm | Σ | >35 µm | 35–5 µm | <5 µm | Σ |
| Salinity | −0.04 | −0.19 | −0.16 | −0.16 | −0.37 | −0.58 | −0.66 | −0.65 |
| | (0.76) | (0.15) | (0.25) | (0.25) | (**<0.01**) | (**<0.01**) | (**<0.01**) | (**<0.01**) |
| Temperature | 0.00 | 0.13 | 0.03 | 0.05 | 0.15 | 0.21 | 0.21 | 0.23 |
| | (0.99) | (0.33) | (0.81) | (0.70) | (0.27) | (0.12) | (0.11) | (0.09) |
| Turbidity | −0.13 | 0.42 | 0.29 | 0.36 | 0.37 | 0.69 | 0.59 | 0.59 |
| | (0.21) | (**<0.01**) | (**<0.01**) | (**<0.01**) | (**<0.01**) | (**<0.01**) | (**<0.01**) | (**<0.01**) |
| $NO_x$[b] | 0.08 | −0.26 | −0.01 | −0.09 | −0.19 | −0.32 | −0.24 | −0.28 |
| | (0.55) | (0.05)[c] | (0.92) | (0.51) | (0.16) | (**0.01**) | (0.07) | (**0.04**) |
| DRP[d] | −0.13 | −0.15 | −0.12 | −0.10 | −0.13 | −0.04 | −0.12 | −0.10 |
| | (0.35) | (0.28) | (0.38) | (0.48) | (0.35) | (0.78) | (0.38) | (0.46) |
| Ammonia[e] | 0.05 | 0.26 | 0.16 | 0.22 | 0.00 | 0.09 | 0.03 | 0.03 |
| | (0.69) | (0.05)[c] | (0.25) | (0.10) | (0.98) | (0.50) | (0.80) | (0.80) |
| Alkalinity | 0.21 | −0.20 | 0.34 | 0.35 | −0.21 | −0.20 | −0.25 | −0.23 |
| | (0.11) | (0.14) | (**0.01**) | (**0.01**) | (0.12) | (0.14) | (0.06) | (0.09) |
| Euphotic zone depth | 0.15 | −0.45 | −0.31 | −0.35 | −0.36 | −0.64 | −0.59 | −0.58 |
| | (0.28) | (**<0.01**) | (**0.02**) | (**<0.01**) | (**<0.01**) | (**<0.01**) | (**<0.01**) | (**<0.01**) |

[a]Significant correlations (P < 0.05) are bolded.
[b]$NO_x$ includes nitrate +nitrite; LOD value (0.02 mg/L) was used as sample data for samples that fell below LOD.
[c]P- values were rounded, hence some 0.05 reported are not significant.
[d]Dissolved Reactive Phosphorus; LOD value (0.005 mg/L) was used as sample data for samples that fell below LOD.
[e]LOD value (0.09 mg/L) was used as sample data for samples that exceeded LOD.

**TABLE 4** Spearman's ρ (P values[a]) between phytoplankton groups and combined *Vibrio* spp. abundances

| Group | Σ *V. parahaemolyticus* | Σ *V. vulnificus* |
|---|---|---|
| *Akashiwo sanguinea* | −0.02 (0.92) | 0.51 (**<0.01**) |
| *Ceratium* spp. | 0.15 (0.31) | −0.09 (0.54) |
| *Dinophysis* spp. | −0.00 (0.98) | −0.23 (0.10) |
| *Heterocapsa* spp. | −0.08 (0.60) | 0.42 (**<0.01**) |
| *Polykrikos kofoidii* | 0.06 (0.71) | 0.18 (0.20) |
| *Prorocentrum* spp. | −0.24 (0.09) | −0.44 (**<0.01**) |
| *Protoperidineum* spp. | 0.07 (0.65) | 0.15 (0.30) |
| *Pseudo-nitzschia* spp. | −0.33 (**0.02**) | −0.54 (**<0.01**) |

[a]Significant P values (P < 0.05) are bolded.

model, the N-S wind component was the only significant predictor of abundance (Tables 6 and 8).

## NMDS and PERMANOVA

After plotting the NMDS coordinates for each *Vibrio*/phytoplankton community sampled, no defined clusters were identified (Fig. 6a). When species were plotted on top of the NMDS coordinates, species with higher salinity tolerances were grouped on the right half of the plot, whereas species with lower preferred salinities were grouped on the left side of the plot. These preliminary trends were confirmed by overlaying structuring variables on top of the NMDS plot as vectors. Significant structuring variables for these communities across site differences included temperature, salinity, euphotic depth, turbidity, and alkalinity (all P values ≤ 0.02). Higher temperatures pull communities toward the top right quadrant of the NMDS ordination, higher salinities pull communities toward the bottom right quadrant, and higher turbidity pulls communities toward the top left quadrant; significant factors were the longest vectors (Fig. 6b).

## DISCUSSION

The interactions among processes in the EMSS provide a complex backdrop for understanding *Vibrio* spp. dynamics. Fluvial input modulates an interrelated suite of hydrographic parameters (18); by affecting cellular processes (e.g., osmotic regulation and photosynthesis), salinity and turbidity likely create biophysical gradients that structure planktonic communities locally (42–44).

This study provides a novel approach to identifying regional factors correlated with *V. parahaemolyticus* and *V. vulnificus* by investigating potential meteorological drivers, association with ecologically relevant particle size fractions, and co-occurrence with harmful algal abundances. Depth-integrated samples were collected exclusively by boat, which reduced potential sampling bias introduced by man-made structures like docks (15, 22, 50–52) and natural stratification of the water column (19, 20). Moving from large- to small-scale processes (physics, hydrology, biology), we explain how these may modulate *Vibrio* spp. abundances.

**TABLE 5** Spearman's ρ (P values[a]) between phytoplankton abundances, hydrographic variables, and nutrients

| Hydrographic variable | *Akashiwo sanguinea* | *Heterocapsa* spp. | *Prorocentrum* spp. | *Pseudo-nitzschia* spp. |
|---|---|---|---|---|
| Salinity | −0.63 (**<0.01**) | −0.53 (**<0.01**) | 0.44 (**<0.01**) | 0.51 (**<0.01**) |
| Temperature | 0.16 (0.26) | 0.05 (0.71) | 0.23 (0.11) | −0.13 (0.38) |
| Turbidity | 0.31 (**0.03**) | 0.29 (**0.05**) | −0.22 (0.15) | −0.53 (**<0.01**) |
| NO$_x$ | −0.22 (0.13) | −0.20 (0.16) | −0.09 (0.52) | 0.27 (0.06) |
| DRP | −0.13 (0.35) | −0.09 (0.52) | −0.16 (0.25) | −0.14 (0.33) |

[a]Significant P values (P < 0.05) are bolded.

**TABLE 6** Fixed effects and significant parameters[a] of best-fit linear mixed effects models for *Vibrio* spp. $R^2$ values for fixed effects ($R^2_{fe}$) and for random effects ($R^2_{re}$) are reported in the last two columns.

| Model[b] | Salinity | N-S wind scalar | DRP | NH₃ | $R^2_{fe}$[c] | $R^2_{re}$[d] |
|---|---|---|---|---|---|---|
| *V. parahaemolyticus* | * | ** | *** | *** | 0.16 | 0.41 |
| *V. vulnificus* | | ** | | | ~ 0 | ~ 1 |

[a]Asterisks indicate significant predictors, with triple asterisks indicating ($P \sim 0$), double ($P \sim 0.001$), and single ($P \sim 0.01$).
[b]Best fit models for each Vibrio spp. were evaluated using temperature, salinity, N-S wind scalar, E-W wind scalar, euphotic zone, turbidity, nitrate +nitrite, dissolved reactive phosphorus, chlorophyll a, alkalinity, total suspended solids, and ammonia (NH3) parameters. All parameters except alkalinity were retained in at least one best fit model; chlorophyll a was only retained in the V. vulnificus model.
[c]Fixed effects (model parameters).
[d]Random effects (site).

## Meteorological factors and *Vibrio* spp.

At the broadest spatial scale, our results suggest the importance of wind as a predictor and/or environmental correlate for *Vibrio* spp. abundances in the EMSS. South winds historically prevail (e.g., 22), whereas west (east) winds can promote regional upwelling (downwelling) over the shelf via Ekman transport (Fig. 7) offshore (inshore toward coast). Shelf transport impacts estuarine conditions by modulating the exchange through tidal inlets (53, 54). Given the EMSS is shallow, even short-duration (24–48 hour) wind events can promote local upwelling and downwelling (55). The observed decreases in *Vibrio* spp. abundances with east winds suggest that advective processes impact abundances through increased salinity and decreased turbidity via influx of marine water at the surface, or via reduction in sediment resuspension from local downwelling. The wind-event duration may affect the strength of the correlation to vibrios. Overall, our data suggests that regional short-term wind patterns (1–24 hours prior to sampling) may be useful metrics to include for modeling *Vibrio* spp. abundances.

## Hydrographic factors and *Vibrio* spp.

Temperature is a principal correlative factor for *Vibrio* spp. abundances (15, 29, 56). Our study temperatures were consistently favorable for *V. parahaemolyticus* and *V. vulnificus* (21, 22, 57), which explains the lack of observed correlation between abundances and temperature. Sampling under low-temperature variability allowed us to identify other driving hydrographic factors in vibrio dynamics.

In the EMSS, high precipitation and river-flow rates are common. The duration of high river discharge in 2019, coupled with historic heavy rainfall events throughout the

**TABLE 7** Best fit LME model output for *Vibrio parahaemolyticus* abundance

| | Random effects | | | |
|---|---|---|---|---|
| | Intercept | | Residual | |
| Standard Deviation | 0.093 | | 0.096 | |
| | Fixed Effects (DF = 30) | | | |
| | Value | Standard Error | *t* value | *P* value |
| (Intercept) | 0.466 | 0.075 | 6.213 | 0.000 |
| Ammonia | 2.745 | 0.714 | 3.845 | 0.001 |
| DRP | 0.101 | 0.018 | 5.736 | 0.000 |
| NOₓ | 0.005 | 0.003 | 1.346 | 0.189 |
| N-S wind scalar | 0.007 | 0.002 | 2.938 | 0.006 |
| E-W wind scalar | 0.001 | 0.004 | 0.231 | 0.819 |
| Salinity | −2.790e-5 | 1.280e-5 | 2.188 | 0.037 |
| TSS | −8.800e-6 | 1.180e-5 | −0.741 | 0.465 |
| Turbidity | 0.002 | 0.002 | 1.112 | 0.275 |
| Euphotic zone | −0.001 | 0.003 | −0.370 | 0.714 |
| Temperature | 0.003 | 0.002 | 1.294 | 0.206 |

**TABLE 8** Best fit LME model output for *Vibrio vulnificus* abundance

| | Random effects | | |
|---|---|---|---|
| | **Intercept** | **Residual** | |
| Standard Deviation | 1.000 | 2.070e-16 | |
| | **Fixed Effects (DF = 29)** | | |
| | **Value** | **Standard Error** | ***t* value** | ***P* value** |
| (Intercept) | −3.276e-16 | 2.574e-16 | −1.273 | 0.213 |
| Ammonia | 2.474e-15 | 1.700e-15 | 1.455 | 0.156 |
| DRP | −8.570e-16 | 9.185e-16 | −0.933 | 0.359 |
| $NO_x$ | 9.337e-18 | 1.117e-17 | 0.836 | 0.410 |
| N-S wind scalar | −1.734e-17 | 5.586e-18 | −3.104 | 0.004 |
| E-W wind scalar | 4.707e-18 | 2.314e-17 | 0.203 | 0.840 |
| Salinity | 7.022e-19 | 3.955e-18 | 0.178 | 0.860 |
| TSS | 6.799e-18 | 4.432e-18 | 1.534 | 0.136 |
| Turbidity | −2.068e-19 | 6.791e-18 | −0.030 | 0.976 |
| Chlorophyll a | −1.136e-17 | 7.870e-18 | −1.443 | 0.160 |
| Euphotic zone | −5.700e-21 | 7.000e-21 | −0.815 | 0.422 |
| Temperature | −1.300e-21 | 1.900e-21 | −0.695 | 0.493 |

central and eastern United States (58), prompted the unprecedented double openings of the Bonnet Carré spillway in Lake Pontchartrain, LA (~200 km west of our study region) and led to extended periods of low salinity conditions in the study region. The combined effects of local and adjacent fluvial inputs represented an ~80% increase in freshwater, relative to typical spring conditions (Dzwonkowski, unpublished data). Although 2019 may represent an anomalous year, these freshwater patterns are predicted to become more common as climate change progresses (59); therefore, our data may provide a glimpse into future conditions. In subtropic estuaries, like the EMSS, salinity can be a stronger structuring variable for *Vibrio* spp. populations than temperature (14). Due to the high influx of freshwater, salinity throughout the coastal bays and central sound sites in the EMSS remained <15 ppt during May and June (Fig. 3). This likely affected the structuring of biophysical gradients within the EMSS and, in turn, the abundances of *Vibrio* spp. within the sampling region.

Local variability of vibrio abundances in response to environmental variables has been demonstrated for *V. cholerae* in Mobile Bay (47) and *V. vulnificus* and *V. parahaemolyticus* (21) in the broader EMSS. *V. parahaemolyticus* abundances did not trend with salinity in our study, which may reflect the species' ability to thrive among a range of salinities (e.g., 23). Low salinities in the early months of the sampling period may have provided more favorable conditions for *V. vulnificus* throughout coastal bays and the EMSS, a preference that has been documented in the GOM (e.g., Galveston Bay, 13) and mid-Atlantic (e.g., Barnegat Bay, 15) regions in the United States. Patterns with salinity are not universal among *Vibrio* spp. or ecological systems; salinity was not a significant predictor of *V. parahaemolyticus* in water when compared across sites on the west (e.g., Washington) and east (e.g., Maryland) coasts of the United States and the northern GOM sectors (60), implying some regional specificity for *Vibrio* spp.

Both *V. parahaemolyticus* and *V. vulnificus* abundances positively correlated with turbidity. Such trends have been previously reported locally (21, 22) and in the broader Southeastern United States (e.g., North Carolina estuaries, 61). Turbidity is mostly caused by resuspension of sediment particles into the water column and can include biological and/or detrital particles. Sediments have been shown to be an important reservoir of vibrios, especially for *V. parahaemolyticus* (21). *V. vulnificus* has also been isolated from sediments, but not at consistently high concentrations (21, 62, 63). Sediment resuspension may represent a pathway for the reintroduction of vibrios into the water column, where they can interact with components of the microbial loop, colonize planktonic substrate, and be consumed by higher trophic level organisms.

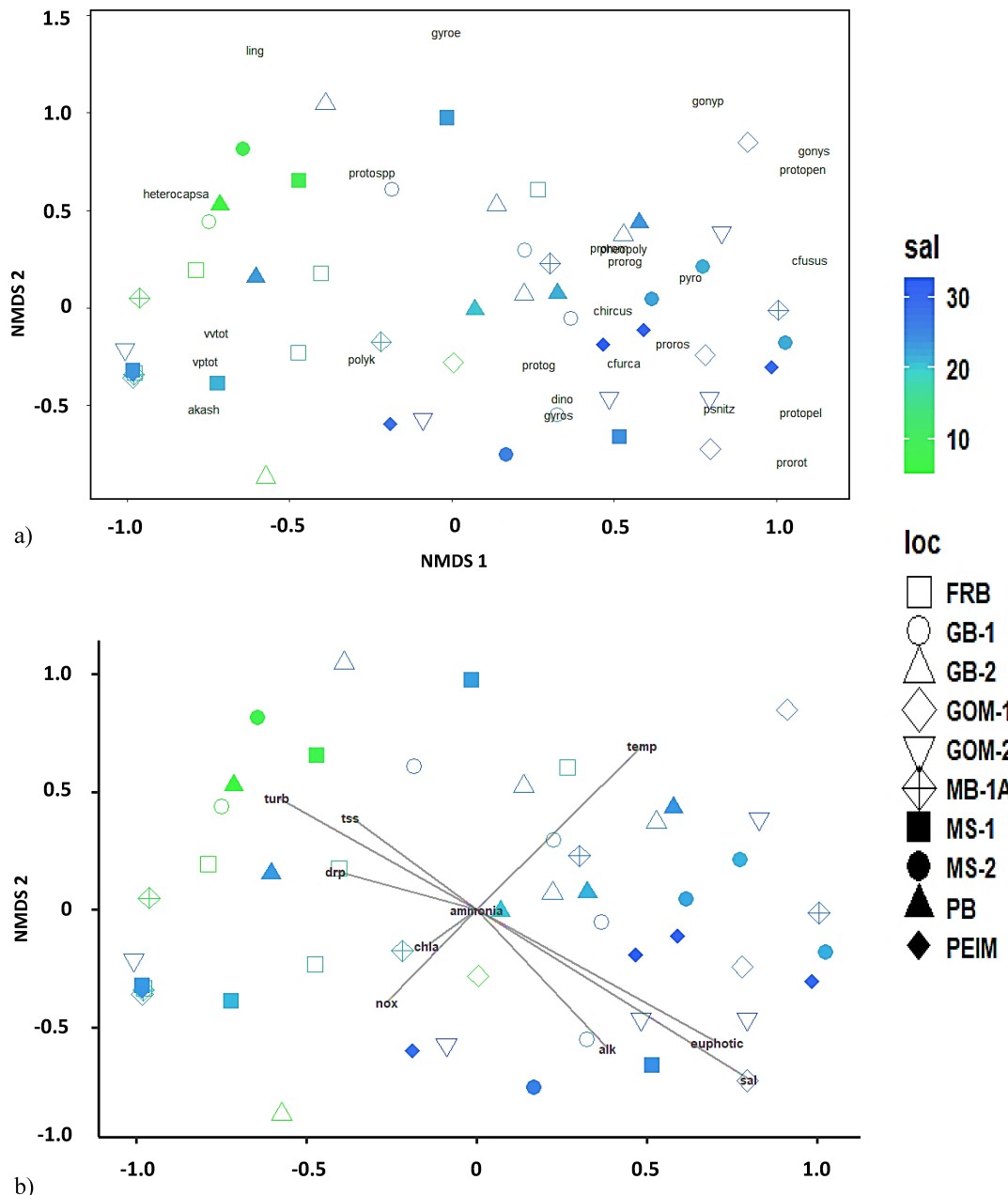

**FIG 6** NMDS plots of planktonic communities in the EMSS. (a) NMDS plot for *Vibrio* spp. and monitored phytoplankton species. Site is indicated by marker shape and salinity is indicated by marker color. Species abbreviations are overlain in black text. Species abbreviations are as follows: akash = *Akashiwo sanguinea*; cfurca = *Ceratium furca*; cfusus = *Ceratium fusus*; chircus = *Ceratium hircus*; dino = *Dinophysis* spp.; gonyp = *Gonyaulax polygramma*; gonys = *Gonyaulax spinifera*; gyroe = *Gyrodinium estuariale*; gyros = *Gyrodinium spirale*; heterocapsa = *Heterocapsa* spp.; ling = *Lingulodinium polyedrum*; pheopoly = *Pheopolykrikos hartmanii*; polyk = *Polykrikos kofoidii*; prorog = *Prorocentrum gracile*; prorom = *Prorocentrum micans*; proros = *Prorocentrum scutellum*; prorot = *Prorocentrum triestinum*; protog = *Protoperidinium grande*; protopel = *Protoperidinium pellucidum*; protopen = *Protoperidinium pentagonum*; protospp = *Protoperidinium* spp.; psnitz = *Psuedo-nitzschia* spp.; pyro = *Pyrodinium bahamense*; vptot = *Vibrio parahaemolyticus* (total); vvtot = *Vibrio vulnificus* (total). (b) NMDS plot with structuring variables overlain as vectors. Vector length indicates the strength of the association between the variable and the observed communities. Variable names/abbreviations are printed at the end of their respective vector line segments. Variable abbreviations are as follows: alk = alkalinity; ammonia = ammonia concentration; chla = chlorophyll a; drp = dissolved reactive phosphorus; euphotic = euphotic zone depth; nox = nitrate/nitrite ($NO_x$); sal = salinity; temp = temperature; tss = total suspended solids; turb = turbidity.

On average, *V. parahaemolyticus* abundances were greatest at FRB, MB-1A, and PEIM, despite these sites having different salinities and turbidity regimes [Tukey's post-hoc

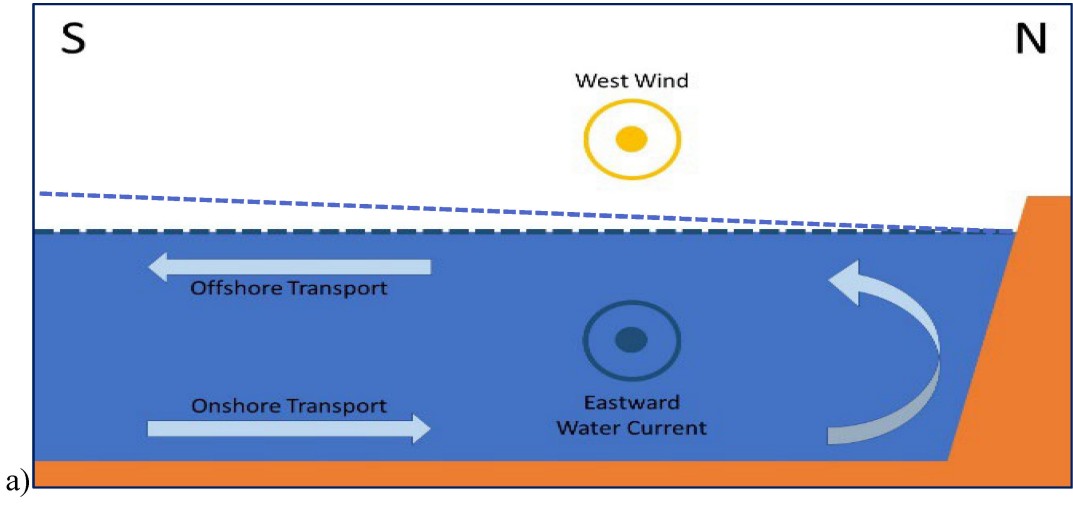

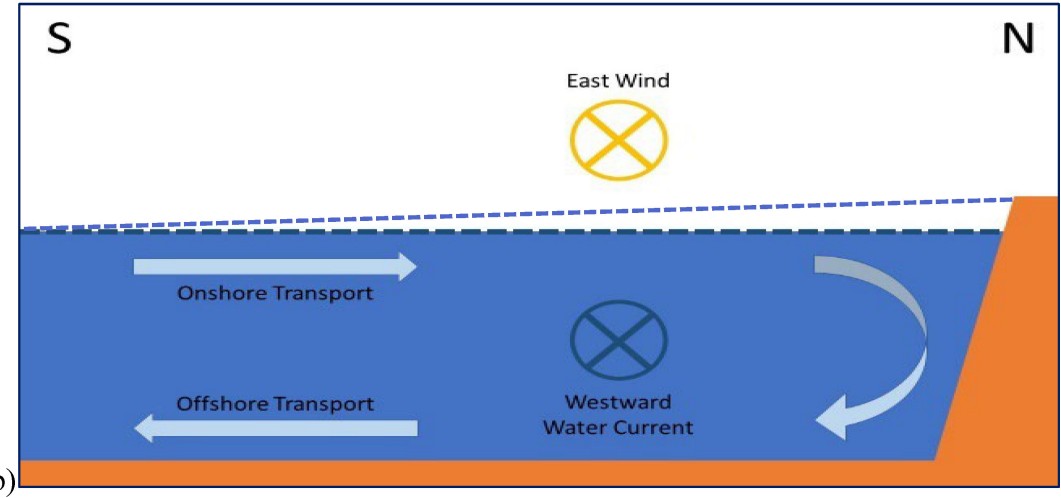

**FIG 7** Wind-driven upwelling and downwelling schematics for the EMSS. (a) West Wind Scenario: these conditions favor the formation of a local upwelling zone. The circle with a dot in the middle indicates wind and water movement out of the page. Although depth-averaged currents are parallel to the wind (blue circle with dot), net surface water transport moves 90° to the right of the direction of wind forcing (Ekman transport), pushing water offshore. The difference in surface water height is denoted by the light blue-dashed line. (b) East Wind Scenario: these conditions favor the formation of a local downwelling zone. The circle with an x in the middle indicates wind and water movement into the page. Although depth-averaged currents are parallel to the wind, net surface water transport moves 90° to the right of the direction of wind forcing (Ekman transport), pushing water toward the shore. The difference in surface water height is denoted by the light blue-dashed line.

(salinity): $P$(FRB:PEIM) =0.003, $P$(MB-1A:PEIM) =0.016; Tukey's post-hoc (turbidity): $P$(FRB:PEIM) =0.051]. Because salinity and turbidity are linked in the EMSS, fluvial input into the system likely affects vibrio levels through multiple mechanisms (lowering salinity and resuspending sediment) that impact distinct reservoirs of *Vibrio* spp. High *V. parahaemolyticus* abundances were found at FRB and MB-1A when turbidity was >10 NTU, whereas high abundances were found at PEIM when turbidity was <5 NTU. We hypothesize that high abundances at the three aforementioned sites may represent two distinct lifestyles of *V. parahaemolyticus*: those that sequester in sediments (occasionally resuspended) and those associated with pelagic plankton (responding to fluvial discharge). If accurate, this suggests a complex interaction between salinity and sediment resuspension in modulating amounts of *Vibrio* spp. in the water column.

## Biological factors and *Vibrio* spp.

The planktonic communities that *Vibrio* spp. inhabit may also affect their population dynamics. Prior work has shown that vibrios readily associate with zooplankton (29, 30, 64, 65), but planktonic relationships are poorly understood regionally. Our results identify correlations with numerically rare harmful algal species actively monitored by a state government agency (i.e., ADPH); neither *Vibrio* spp. nor these harmful species correlated with metrics of the bulk phytoplankton (i.e., chlorophyll-a). The lack of correlation suggests that these harmful algal species are not the main drivers of chlorophyll biomass in the region and/or that there is a considerable detrital chlorophyll signal in the EMSS—potentially from the benthos (66, 67). Furthermore, these data show that chlorophyll-a is a poor predictive metric for *V. parahaemolyticus* and *V. vulnificus* regionally in the warmer seasons. Past studies in the northern GOM also found no significant relationships between *Vibrio* spp. and chlorophyll-a (22, 60), opposite of studies in other estuarine systems: Barnegat Bay, NJ (15); Venetian Lagoon, Italy (68); and Great Bay, NH (69). Broadly, these data suggest that bulk chlorophyll-a is likely a regionally specific variable for vibrio correlation that may be confounded by other physical and/or hydrographic factors.

In our study sites, 30–50% of target vibrios were associated with particles ≥5 µm. The monitored harmful algal groups varied in size from 2 to 230 µm (Table S1), demonstrating diversity in the size of biological particles available for *Vibrio* spp. to associate with. Sediment particles also vary in size, from clay grains (≤2 µm) to coarse sand (up to 1 mm). Detrital organic particles (e.g., aggregates and marine snow) can be even larger, and *Vibrio* spp., like other bacteria, may colonize these surfaces as a biofilm (70). In North Carolina estuaries, 3–60 µm particulates in the water column (attributed to phytoplankton) were associated with an increase in particle-associated *Vibrio* spp. abundance (71); the frequency of particle association also decreased with increasing salinity. Although *Vibrio* spp. interactions with specific particle-size groupings were not isolated by Hsieh et al. (71), an opposite trend between salinity and *V. vulnificus* particle size association was seen in the EMSS (Fig. 5a).

In this study, *V. vulnificus* was associated with smaller particles in lower salinities, suggesting a potential role for freshwater input in facilitating particle association. As *V. vulnificus* prefers low salinities (5–15 ppt), associating with large particles in high salinity may offer proximity to cellular exudates (food), potential protection from osmotic stress (e.g., availability of leaky osmolytes; 72), and/or protection from consumption by protozoan grazers. Conversely, *V. parahaemolyticus* was associated with large particles over a greater range of salinities than *V. vulnificus*. Kaneko and Colwell (73) demonstrated that *V. parahaemolyticus* more readily associated with copepods in lower (2 ppt) salinities (compared with 16 ppt) and that adherence to chitin-based organisms may offer protection from thermal and osmotic stressors. Our data are consistent with the ideas that *V. vulnificus* may use particle associations for adapting to unfavorable conditions (e.g., higher salinities), whereas *V. parahaemolyticus* may be better adapted among the range of salinities observed (i.e., does not consistently require associations with larger particles). The size of particles that vibrios associate with can have notable implications for retention within microbial-loop processes (e.g., viral lysis and ingestion by microzooplankton) or potential shunting to higher trophic levels via ingestion by larger organisms like oysters, crabs, and fish (74).

In addition to particle-size associations, correlations between summed target vibrio abundances and specific phytoplankton groups were identified. *V. vulnificus* abundances were positively correlated to low salinity-preferring phytoplankton (*A. sanguinea* and *Heterocapsa* spp.) and negatively correlated to species with higher salinity tolerances (*Prorocentrum* spp. and *Pseudo-nitzschia* spp.). *V. parahaemolyticus* was only negatively correlated with *Pseudo-nitzschia* spp. It is important to note that these correlations describe environmental associations, not physical attachment or association with the algal phycosphere. Positive correlations can infer environmental conditions where both species thrive. In PERMANOVA-NMDS analysis, communities characterized by the

presence of *A. sanguinea* and *Heterocapsa* spp. were driven by low salinity, shallow euphotic depth, and higher turbidity (i.e., freshwater discharge dominated), whereas communities dominated by *Pseudo-nitzschia* spp. were driven by high salinity, deep euphotic zone, and low turbidity (i.e. proportionally higher effect of GOM water).

## Overall drivers of vibrio abundance

Although salinity, turbidity, and the E-W wind component were identified as significant correlates in the correlation analysis, they did not remain significant parameters of the LME model for either vibrio species. Differences in site accounted for a large proportion of *V. parahaemolyticus* variation (41%) and nearly 100% of *V. vulnificus* variation. Therefore, significant variables identified by correlations are likely intrinsically tied to site. This is unsurprising, as proximity to freshwater sources/outflows and effects of the E-W wind component (i.e., along-shelf wind regionally) underlie site-specific differences. Locally, the E-W wind scalar may affect turbidity, wave action, and physical mixing potential as a function of the water column depth and the fetch length of open water that the wind can act upon; water depth and fetch length varied among sites. Interestingly, the N-S wind scalar was retained in the model despite not being significant in univariate analyses; this could be attributed to its mechanistic effect on advective processes, which move water (and its components) into and out of the estuary. Ammonia was also retained, but this is likely due to the lack of variation—an artifact of the analysis method. The fixed effects in the model accounted for relatively little variation in *V. parahaemolyticus* abundance (16%) and practically none of the variation in *V. vulnificus* abundance.

## Future research/monitoring of *Vibrio* spp. and implications

Our findings highlight the potential benefit of expanding real-time monitoring systems along coastal margins. Estuarine conditions are dynamic temporally and spatially; thus, more frequent sampling can provide stakeholders with better information regarding potential conditions favoring increases in *Vibrio* spp. abundance. Given the *Vibrio* spp.-plankton correlations identified here, we suggest the possibility that existing phytoplankton monitoring programs be explored and/or leveraged as an early warning/alert tool for additional bacteriological sampling. This knowledge could justify future investment in autonomous phytoplankton imaging systems (e.g., 75), which can detect certain harmful algal species that strongly correlate (positively and negatively) to *Vibrio* spp. in near real time. Past *Vibrio* spp. studies in the northern GOM have focused on pathogenic-strain population dynamics and predicting impacts on public health through shellfish vectors; elucidating the complex interactions between meteorological, hydrographic, and biogeochemical processes that underlie *Vibrio* spp. abundances in this region may help inform the next iteration of *Vibrio* spp. risk models (76–78).

Although individual estuary systems are unique, the findings of this study may be applicable to other coastal margin systems. In particular, shorter estuaries—which can be strongly influenced by upwelling and downwelling events—are more likely to experience similar physical forcing mechanisms as observed in the EMSS. For example, shallow, fluvially dominated microtidal regions like the northern GOM and tributaries of the Pamlico Sound System (North Carolina, USA) may display similar hydrographic trends and subsequent *Vibrio* spp. correlations as those identified in this study.

Climate models for the GOM coast indicate that extreme precipitation events are predicted to increase during later decades of this century (59). As the climate warms, the hydrological cycle intensifies, leading to more intense precipitation events (79). Greater rainfall in coastal areas and river basins draining to the northern GOM will likely impact the duration of low salinity waters in estuarine margins.

As coastal freshwater flooding becomes more frequent with climate change, conditions favorable for the proliferation of planktonic *V. vulnificus* could become more common; subsequently, coastal planktonic community composition may shift to predominately low-salinity and high-turbidity tolerant species. Our study suggests that

low salinities also affect vibrio-particle interactions. These relationship changes could have implications for increased assimilation of *Vibrio* spp. into economically important organisms. Larval oysters have been shown to derive up to 60% of their food from particles between 0.5 and 10 µm (80), and adult oysters can concentrate bacterioplankton in their tissues by ~100 fold relative to ambient levels in the water column. Association with small particle sizes in nearshore localities may reduce the amount of particle-associated *V. vulnificus* directly ingested by higher trophic-level organisms, especially those that are the basis of coastal fisheries (shrimp, crabs, anchovies, etc.). Conversely, if *V. parahaemolyticus* associates with larger particles (e.g., chitinous) in low salinities to adapt to stressful osmotic conditions (73), they may be more easily consumed by larger predators (fish, blue crabs, and shrimp) and assimilated into gut microbiota. Thus, future environmental changes may affect the abundance and assemblage of *Vibrio* spp. in certain reservoirs, ultimately affecting which vectors contain the greatest vibrio risk.

## MATERIALS AND METHODS

### Sampling sites

This study was conducted from April to October of 2019 in the EMSS and coastal Alabama, both under the hydrographic influence of Mobile Bay. Study sites included coastal bays (Fowl River Bay, FRB; Grand Bay, GB-1, GB-2; and Portersville Bay, PB), central sound (Mississippi Sound, GOM-1, MS-1, MS-2; and Western Mobile Bay, MB-1A), and barrier island localities (Pelican Island, PEIM; and Petit Bois Pass/Gulf of Mexico, GOM-2) (Fig. 1). Sites were expected to display fluctuations in salinity throughout the year due to freshwater input. Average water depths at each site can be found in Table S2.

### Meteorological data

Wind speed, wind direction, and precipitation for 2019 were recorded by the ARCOS Meteorological Station on Dauphin Island (DI). ARCOS stations closer to sampling points in the EMSS were considered (Fig. 1, purple stars), but trends in wind speed and direction at these sites were highly correlated to the DI station (r > 0.69, r-critical = 0.17 at α = 0.01; data not shown), and the DI station had the most consistent data quality during the study period (48). Archived data were accessed from the ARCOS website (https://arcos.disl.org/). Wind direction measurements were recorded at 10 m above ground in standard meteorological notation (360°/0° signifying North). Wind speed and direction were transformed into scalar components, N-S and E-W (81). Tidally filtered river discharge data for the Mobile River (a key tributary to the Mobile-Tensaw Delta) were collected by the United States Geological Survey (USGS) river gauge station #02470629 in Bucks, AL. Archived data were accessed from the USGS data repository (https://waterdata.usgs.gov). ARCOS station precipitation data were used as a proxy for local freshwater input via precipitation, whereas freshwater input from upstream precipitation was captured by river discharge data.

### Field sampling

Samples for *Vibrio* spp. enumeration were collected monthly at each site in conjunction with the Alabama Department of Environmental Management (ADEM) water quality monitoring field team. Samples were collected alongside standard water quality monitoring efforts conducted by the Alabama Department of Environmental Management. ADEM is the primary public entity charged with evaluating water quality trends in the state; therefore, their monitoring methods and schema are well established and hold regulatory authority, and their data are made publicly accessible. ADEM begins sampling in April and concludes in October, following the pattern of sampling when the water is warmest, production is highest, and water usage by the public is most common.

The euphotic zone depth at each site was determined using an LI-400 photosynthetically active radiation meter with a LI-192 underwater quantum sensor (LiCor Biosciences, Lincoln, NE); the base of the euphotic zone was defined as the depth where less than 1% of ambient surface photosynthetically active radiation was detected. Full-depth profiles for water temperature, conductivity (salinity), pressure (depth), and total dissolved solids were recorded using a YSI EXO 2 data sonde (YSI/Xylem Inc, Yellow Springs, OH) with an approximate 0.5–1 meter resolution. Accordingly, depth-integrated euphotic zone samples were collected using a sump pump (3028 L/h) attached to a hose that was raised and lowered through the water column at approximately 1 meter every 5 seconds. Two 4 L replicate water samples were collected at each site in sterile polypropylene bottles, and an additional 1 L sample was collected for phytoplankton analysis and stored in a 1 L glass jar preloaded with 7 mL of 12% Lugol's iodine solution fixative. Hydrographic/nutrient parameters (e.g., alkalinity, turbidity, nitrate, nitrite, phosphate, and ammonia) and particulates (e.g., chlorophyll-a and total suspended solids) were assessed by ADEM using their standard operating procedures (82). All sampling was conducted within a 3 hour window of 07:00 local time (GMT - 6 hour standard time). Water samples (from the 4 L replicate bottles) for *Vibrio* spp. enumeration were placed in a cooler, transported to the United States Food and Drug Administration Gulf Coast Seafood Laboratory (Dauphin Island, AL, USA) within 40 min of returning to the dock, and were processed on the same day. Hydrographic and nutrient samples collected by ADEM were transported on ice and delivered to the ADEM Chemistry and Microbiology Lab (Mobile, AL, USA). Chlorophyll-a samples were frozen until analysis.

## *Vibrio* spp. analysis

Samples were processed using sequential filtration to fractionate the planktonic community based on size. Duplicate 100 mL aliquots from each sample were sequentially filtered through 35 µm Nitex mesh (BioDesign Inc., Carmel, NY) and then a 5 µm polycarbonate membrane (Isopore, Darmstadt, Germany), with particles smaller than 5 µm being pelleted via centrifugation (10 min at 5000 x $g$). These size thresholds were utilized to roughly fractionate the planktonic constituents by trophic guild ( >35 µm =zooplankton and large microphytoplankton; 35 µm −5 µm = microzooplankton, micro- and nano-phytoplankton; <5 µm =picophytoplankton, nanozooplankton, and bacteria). Each filter and pellet were vortexed for 1 min in 10 mL of phosphate-buffered saline (PBS; 0.765% NaCl, 0.072% $Na_2HPO_4$, 0.021% $KH_2PO_4$, pH 7.4 ± 0.2). The resuspensions were serially diluted in PBS and used as inoculum for a three-tube Most Probable Number (MPN) assay for each size fraction (>35 µm, 35 µm—5 µm, and <5 µm) in alkaline peptone water (83). Following overnight enrichment, crude cell lysates were prepared by heating aliquots from tubes with positive growth in a heat block at 95–100°C for 10 minutes (previously described, 84) and stored at −20°C until used in real-time PCR; all thawed lysates were centrifuged at 10,000 × $g$ for 2 min before using as templates for real-time PCR. *V. parahaemolyticus* (*tlh*) and *V. vulnificus* (*vvh*) detection assays included an internal amplification control and were conducted on the AB 7500 Fast (Life Technologies, Foster City, CA) as previously described (85). Additional details about primers, reaction components, and cycling parameters can be found in Table S3.

## Select phytoplankton and harmful algal species' identification

Preserved 1 L phytoplankton samples were processed by the Alabama Department of Public Health (ADPH) Phytoplankton Unit. The ADPH lab primarily monitors for larger dinoflagellates and harmful algal bloom-forming species in coastal waters; therefore, not all phytoplankton groups were identified and counted. Target genera cells in a representative sample aliquot were visually identified and enumerated using light microscopy. A concentration factor of 103 was used to scale subsample cell density to estimate cell density/L (44, 86).

## Statistical analysis

MPN values were determined for *V. parahaemolyticus* and *V. vulnificus* in each size fraction of each sample using a standard MPN table (87) and then log-transformed; these values were then averaged for each site. Samples non-detectable in all MPN tubes were considered below the limit of detection (<30 MPN/L). For averaging, these samples were assigned a value of 15 MPN/L (1.18 log MPN/L). Combined, or summed, *Vibrio* spp. abundances were determined by summing the three average vibrio abundances for each size fraction.

Spearman's non-parametric rank-based correlations were conducted to determine if any monotonic relationships existed between meteorological parameters (wind direction, wind speed, wind vectors), hydrographic variables [euphotic zone averages: temperature, salinity, turbidity, nitrate/nitrite ($NO_x$), dissolved reactive phosphorus (DRP), ammonia, chlorophyll-a, alkalinity], biological parameters (phytoplankton species abundances), and *Vibrio* spp. abundances. For samples that fell below or exceeded the LOD for nutrient analyses, the LOD value was used as sample data ($NO_x$: 0.02 mg/L; DRP: 0.005 mg/L; ammonia: 0.09 mg/L) in statistical analyses. Hydrographic parameters were tested against *Vibrio* spp. abundance associated with each individual size fraction (>35 μm, between 35 and 5 μm, <5 μm) and summed *Vibrio* spp. abundances, whereas meteorological and biological parameters were only tested against summed *Vibrio* spp. abundances. Spearman's correlations were used to help refine variables to include in subsequent analyses.

Linear mixed effects (LME) models were calculated to determine significant environmental predictors of *Vibrio* spp. abundances after accounting for site-based variation. LME models were created in R (88) and RStudio (89) using the *nlme* package (90) for summed vibrio abundances. Site was a random effect, with environmental correlates (temperature, salinity, N-S wind component, E-W wind component, euphotic zone depth, turbidity, $NO_x$, DRP, chlorophyll-a, total suspended solids, and ammonia) included in a global model as fixed effects. Correlations between fixed-effect variables were evaluated using the *car* package (91). All iterations of fixed effects within the global model were evaluated using the "dredge" function within the *MuMIn* package (92) to determine the best-fit model. Significant parameters were determined from the best-fit model output, and the output was assessed with the "r.squaredGLMM" function to determine $R^2$ values attributed to fixed effects and random effects in the model. This process was completed for *V. parahaemolyticus* and *V. vulnificus* summed abundances.

PERMANOVA and non-metric NMDS approaches were used to determine environmental variables that structured planktonic communities (i.e., *Vibrio* spp. and harmful algal groups) in the sampling region. These analyses were calculated by using the *vegan* package (93) in R. Environmental correlates were reduced to two dimensions in the NMDS analysis and were plotted using the *ggplot2* package (94). The PERMANOVA was completed using the "adonis" function and was set to run with 999 permutations. Vectors were calculated using the "envfit" function and were overlain onto the NMDS plot.

## ACKNOWLEDGMENTS

B.H. Morrison would like to thank Kyle Halstead, Sydney Acton, Alex Marquez, and Allie Smith for their help with field sample collection, as well as Madison McGough, Keri Lydon, Vicki Pruente, Whitney Neil, and James Kelly for their assistance with processing samples in the lab. The authors would like to thank all anonymous reviewers for their input and suggestions throughout the review process. This work was supported by the Dauphin Island Sea Lab-U.S. Food and Drug Administration Joint Fellowship Program (#5U19FD005923-04) awarded to B.H. Morrison. Field operations were made possible through a collaboration with the Alabama Department of Environmental Management—Coastal Program and were facilitated by Joie Horn and Clark Gerkin. Phytoplankton identification was conducted by Drew Sheehan and the staff of the Alabama Department of Public Health Phytoplankton Lab.

## AUTHOR AFFILIATIONS

[1]Dauphin Island Sea Lab, Dauphin Island, Alabama, USA

[2]Stokes School of Marine and Environmental Sciences, University of South Alabama, Mobile, Alabama, USA

[3]FDA, Division of Seafood Science and Technology, Gulf Coast Seafood Laboratory, Dauphin Island, Alabama, USA

## PRESENT ADDRESS

Blair H. Morrison, Mobile Bay National Estuary Program, Mobile, Alabama, USA

## AUTHOR ORCIDs

Blair H. Morrison http://orcid.org/0000-0002-3784-6056
Jessica L. Jones http://orcid.org/0000-0002-5077-216X
Brian Dzwonkowski http://orcid.org/0000-0002-2333-2185
Jeffrey W. Krause http://orcid.org/0000-0003-2479-6229

## FUNDING

| Funder | Grant(s) | Author(s) |
|---|---|---|
| HHS | U.S. Food and Drug Administration (FDA) | #5U19FD005923-04 | Blair H. Morrison |
| | | Jessica L. Jones |
| | | Jeffrey W. Krause |

## AUTHOR CONTRIBUTIONS

Blair H. Morrison, Conceptualization, Data curation, Formal analysis, Investigation, Methodology, Validation, Visualization, Writing – original draft, Writing – review and editing | Jessica L. Jones, Conceptualization, Funding acquisition, Methodology, Resources, Supervision, Validation, Writing – review and editing | Brian Dzwonkowski, Conceptualization, Methodology, Software, Supervision, Visualization, Writing – review and editing | Jeffrey W. Krause, Conceptualization, Funding acquisition, Investigation, Methodology, Resources, Supervision, Visualization, Writing – review and editing

## DATA AVAILABILITY

Morrison, B., Jones, J., Dzwonkowski, B., & Krause, J., 2023. Tracking Vibrio: Population Dynamics and Community Ecology in Alabama Estuaries (Version 1) [Data set]. Dauphin Island Sea Lab. https://doi.org/10.57778/2X11-XN28.

## ADDITIONAL FILES

The following material is available online.

### Supplemental Material

**Supplemental figures and tables. (Spectrum03674-23-s0001.pdf).** Figures S1-S6; Tables S1-S3.

### Open Peer Review

**PEER REVIEW HISTORY (review-history.pdf).** An accounting of the reviewer comments and feedback.

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
