## [Reviewer comments · Microbiology Spectrum]

Microbiology Spectrum

Tracking *Vibrio*: Population Dynamics and Ecology of *Vibrio parahaemolyticus* and *V. vulnificus* in an Alabama Estuary

Blair Morrison, Jessica Jones, Brian Dzwonkowski, and Jeffrey Krause

Corresponding Author(s): Blair Morrison, Dauphin Island Sea Lab

Review Timeline:

Submission Date:

October 27, 2023

Accepted:

March 5, 2024

Editor: Allison Veach

Reviewer(s): The reviewers have opted to remain anonymous.

Transaction Report:

DOI: <https://doi.org/10.1128/spectrum.03674-23>

Re: Spectrum03674-23 (Tracking *Vibrio*: Population Dynamics and Ecology of *Vibrio parahaemolyticus* and *V. vulnificus* in an Alabama Estuary)

Dear Miss Blair H Morrison:

After careful internal review, it is clear the authors have addressed previous reviewers comments thoroughly. Therefore, I'm happy to inform you your manuscript has been accepted, and I am forwarding it to the ASM production staff for publication. Your paper will first be checked to make sure all elements meet the technical requirements. ASM staff will contact you if anything needs to be revised before copyediting and production can begin. Otherwise, you will be notified when your proofs are ready to be viewed.

Sincerely,
Allison Veach
Editor
Microbiology Spectrum